# Models of Care and Relationships with Care Homes: Cross-Sectional Survey of English General Practices

**DOI:** 10.3390/ijerph192214774

**Published:** 2022-11-10

**Authors:** Krystal Warmoth, Claire Goodman

**Affiliations:** 1Centre for Research in Public Health and Community Care (CRIPACC), University of Hertfordshire, Hatfield AL10 9AB, UK; 2National Institute for Health Research Applied Research Collaboration East of England (NIHR ARC EoE), Cambridge CB2 8AH, UK

**Keywords:** residential care, primary care, nursing homes, GP, long-term care

## Abstract

The Enhanced Health in Care Homes framework for England sets out standards for how primary care should work with care homes. How care home staff and General Practitioners work together and the quality of their working relationships are core to resident healthcare. This study explored the current models of care and relationships between General Practitioners′ practices and care homes. Sixty-seven respondents from 35 practices were recruited in the East of England and completed a semi-structured online survey. Responses were analysed using descriptive statistics. Free text responses were interpreted thematically. The number of care homes that practices supported ranged from 0–15. Most reported having designated General Practitioners working with care homes and a good working relationship. Despite the national rollout of the Enhanced Health in Care Homes framework, two-thirds reported no recent changes in how they worked with care homes. There is a shift towards practices working with fewer care homes and fewer accounts of fractured working relationships, suggesting that residents’ access to primary care is improving. The continuing variability suggests further work is needed to ensure it is driven by context, not inequitable provision. Future work needs to address how policy changes are changing work practices and residents’ health outcomes.

## 1. Introduction

In March 2020, there were an estimated 457,000 care home beds across England [1]. Care homes provide long-term care for older people who are in their last years of life, with complex healthcare needs [2]. Recent policies in England urge primary care to provide collaborative and proactive care to care homes [3,4]. Care home residents rely on primary care physicians or General Practitioners (GPs) for medical care and access to specialist services [5]. Despite an expectation of integration between health and social care organisations, there is a lack of understanding about how care homes and the NHS work together.

The COVID-19 pandemic highlighted how little was known centrally about care home residents, and the support they needed and, the lack of integration between health and social care systems [6,7,8]. Primary care is recognised as key for the provision of healthcare for residents but there is no consensus about how GPs should respond. During the pandemic, it was the pre-existing strength and quality of the relationships between care homes and GPs that determined the care home staff′s capacity to respond to the pandemic [9].

Historically, care homes have worked with multiple practices and GPs, with patterns of working determined by local history and practice [10,11]. These were diverse and included general medical care provided by general practices, community services linked to homes, outreach clinics, care home specialist nurses or support teams, pharmacist-led services, designated NHS hospital beds and enhanced payment schemes for GPs to undertake additional work [10]. Care homes report commonly working with multiple GPs [10,12] and more than 14 different professionals/services (other than GPs) [12]. Not only was there great variation in how health care services and care homes worked together but the frequency of contact [13]. These varied ways of working resulted in unequal access to NHS services and resources for care home residents, particularly expertise that is vital and pertinent (e.g., dementia, rehabilitation, and end-of-life care). A lack of trust and poor working relationships between the primary care and care homes were a barrier to collaboration and integrated working [12,13].

Recent policy changes set out an agenda for primary care services working in Primary Care Networks (PCN) to support the delivery of the ‘Enhanced Health in Care Homes′ framework (EHCH) [4]. The EHCH specifies that each PCN should have a designated care home coordinator, working with a multidisciplinary team to deliver a structured approach for the assessment of new residents and reviewing residents′ ongoing care. It recognises GPs as key in supporting this enhanced delivery for residents [4]. Since its introduction, the experience of GPs and primary care staff delivering healthcare is underexplored [5]. This study explored the current models of care and relationships between GP practices and care homes. The research questions were: What is the most common model of care with care homes used at GP practices?What is the range of models of care used with care homes used at GP practices?Have GP practices changed how they work with care homes in the last year? If so, how?What are the perceptions of the quality of the relationships with care homes?

## 2. Materials and Methods

A semi-structured online survey was sent to GP practices in the East of England during February 2022. Staff were surveyed via Thiscovery (www.thiscovery.org/ accessed on 1 February 2022), an online research platform developed by THIS Institute at the University of Cambridge. A mix of closed and open-ended questions asked about the practice, its current model of care (e.g., designated GP for care home and locality-led care home-specific teams), its relationship with care homes, and any changes experienced in the last year. Respondents were asked about what works well or how GPs and care homes could be supported to improve residents’ access to healthcare. The survey was reviewed by 3 GPs for wording and understanding and then user-tested by 5 independent individuals, before circulation.

The East of England has a population of 6.2 million with a mix of urban and rural, affluent and deprived communities and more than 51,000 care home beds [14]. We recruited via the local Clinical Research Network. The survey was open from February to March 2022. Responses were analysed using descriptive statistics and free text responses were interpreted thematically [15,16].

The study was reviewed and given a favourable opinion by the University of Hertfordshire Ethics Review Board (cHSK/SF/UH/04698) and Health Research Authority (300163).

## 3. Results

Sixty-seven responses were received from 35 general practices, with a response rate of 19% (see Table 1). Respondents were 47 general practitioners, 7 practice nurses, 3 nurse practitioners, 1 practice manager, and 9 other primary care staff.

The number of care homes that individual practices reported working with ranged from 0 to 15. Nearly a third of the practices worked with one care home (29%). Eight respondents (12%) did not know how many care homes their practice supported.

The favoured model of care was each care home had a designated GP (40%) (Table 2). Respondents also described additional roles with care home responsibilities (e.g., Advanced Nurse and Paramedic Practitioners) and multidisciplinary teams. One respondent was unsure about their model of care. Two-thirds of respondents reported that how they worked with care homes had not changed in the last year (66%). However, they reported how the pandemic had led to changes in delivery (e.g., virtual visits instead of in-person), and new staff supporting GPs with care homes. Few respondents mentioned the EHCH specifically, despite describing multidisciplinary teams′ involvement and GPs aligning with care homes. One respondent commented how EHCH had resulted in less funding as it did not consider the work involved to accommodate the “assimilation of new residents” and end-of-life care into existing caseloads. Respondents noted upcoming changes in local services contracts. These were probably linked to EHCH but respondents did not make the connection.

Nearly a quarter (23%) characterised their practice′s relationship with care homes as a good personal relationship, and 20% of respondents said that they used informal as much as formal arrangements (Table 2). Two respondents differentiated between care homes when describing the strength of the working relationship. Only 1 reported having a limited or lack of relationship.

When comparing the models of care with how the relationships were categorised, the practices using the most common model (each care home had a designated GP) generally described their relationship as a good personal relationship (90%), using informal as much as formal arrangements (77%), and efficient working relationship (56%). All 5 practices with locality-led care home-specific teams reported having flexible and responsive 2-way communication, working through informal and formal arrangements, supportive and educative working relationships, good personal relationships, efficient working relationships and good adherence to referral procedures and processes. The agreement across other models varied with none having 100%. The practices who reported participating in Local Enhanced Service for care homes largely stated they had good personal (95%) and efficient working relationships (79%). The only respondent who categorised the relationship negatively reported using the model where most of GPs assimilated care home work into their everyday caseloads.

We reviewed the responses from the same practices to identify discrepancies. Some variabilities were found in the number of care homes, the current model of care and the relationships with care homes. For example, two respondents from the same practice stated a different number of care homes: one stated that they supported six care homes while the other stated four care homes.

## 4. Discussion

Delivering primary care in care homes comes with many challenges. Previous research has reported an enduring history of ad hoc and inequitable approaches, with a great deal of variation in primary care support and provision of services [10,12,17]. The present survey aimed to identify the ways of workings between GP practices and care homes. The most reported model of care with care homes in this survey was each care home had a designated GP. Various models of care used with care homes were also reported by GP practices and included care home specialist teams and additional roles with care home responsibilities. Most respondents did not report any changes in how their practice worked with care homes in the last year despite policy changes. The perceptions of the quality of the relationships with care homes seemed to be overwhelmingly positive.

Our findings suggest a shift towards practices working with fewer care homes when compared to previous studies [10,12]. There was evidence of how national policies (EHCH) and learning from the pandemic had affected the ways practices worked with care homes and a process of rationalisation of practices being aligned with care homes. The most notable change for care homes is that a designated GP per care home or one GP for all the practice′s care homes are implemented as previous studies had found care homes working with up to 30 practices [10]. Some credited this shift to the adoption of EHCH [4], specifically the named GP for each care home, weekly ‘rounds’, and multidisciplinary teams [3,4]. There was, however, variation within the findings that did not appear to be explained by context or respondent characteristics. Ongoing variation in access to and support found in this study mirrors some previous findings [10,12]. These patterns of working may not change easily or need time and resources to enable change [11]. The EHCH is not optional so all practices should have a coordinator and multidisciplinary teams working with the care homes. The findings suggest that there has been some change in how services are organised but it is not universal or uniform.

The rollout of EHCH was brought forward due to the pandemic [18]. The majority reported no changes in the last year although they did cite how the teams worked remotely with care homes and changes to incentive payments. These could be related to the EHCH and pandemic but are indicative of organic and incremental changes to how GPs work with the care home. Perhaps, they provided an additional impetus offering the validation and incentive to work in these ways. A network of healthcare teams along with having a single, designated general practitioner are not completely new ways of working [10] and have been suggested to encourage optimal healthcare provision in care homes [11]. They could also have been encouraged to improve relationships with care homes as those practices who reported participating in Local Enhanced Service for care homes in this study viewed the relationships positively. Few respondents made the link between how the COVID-19 pandemic changed the delivery of care (remote consultations instead of in-person visits) and greater access or continuity of care [19,20]. The challenges of EHCH implementation (e.g., weekly rounds) and additional workload for multidisciplinary team meetings have been reported, suggesting that changes are occurring but not at pace [18,21].

The reports of primary care staff having a good relationship with care homes using a mix of formal (e.g., regular meetings) and informal arrangements (e.g., unscheduled calls) are suggestive of a change in attitudes and how GPs and colleagues work with care home staff. Previous studies found poor working relationships between primary care and care homes and a lack of integrated working [12,13]. Those who reported having designated GPs for care homes and care home-specific teams largely viewed their relationships positively, which could mean their relationships are benefiting from this model.

Little is known about the role of care home staff in how care is organised between the GP and the multidisciplinary team. Learning from the pandemic suggests that it is important that care home staff are actively involved in influencing access to medical care [9]. The financial implications of the change in the services contracts for care homes were also unclear, as GPs are compensated for the number of care home beds in contract arrangements [22]. There may be unintended consequences if GPs who were more flexible in working with care homes now restrict their involvement to what is in the EHCH. How the policy is being interpreted to fit local circumstances needs investigation. For example, there is a trend for increasing care home size and consequently, multiple GPs will be needed due to the greater number of residents [10,12]. Future investigations need to examine the context of the care home and the resident characteristics and how that influences the way the EHCH is delivered.

This survey provides a snapshot of how primary care is currently organised to work with care homes in the East of England. The projected increase in demand for long-term care in the next twenty years [23] requires a coherent and consistent response. It is unlikely that one model of care will be the most effective and may need to be modified to fit the local needs and structures [11,24]. Nevertheless, this survey demonstrates two key findings; there has been an observable shift in the organisation of primary care for this frail population but how this is achieved appears to still be arbitrary. Commissioners need to ensure that care homes currently receive appropriate timely and equitable care and integrated working between primary care and care homes are encouraged.

One strength of the study was the number of respondents that completed the survey despite an ongoing pandemic and high demand on healthcare services. This is also the first study exploring how care is organised and the relationships between GP practices and care homes following the adoption of the EHCH. Survey limitations include sampling in a specific region of England and not being able to demonstrate if these provision changes lead to improved resident and staff outcomes. Respondents may also represent GPs interested in care homes. It is difficult to know what the key components for a good working relationship or the specific elements are that achieved it as only one respondent reported a limited or lack of relationship. The absence of negative comments is remarkable when compared with previous studies [10] and may be attributable to improved relationships due to the pandemic [20] or reflect regional differences in the quality of relationships. The key features of a good primary care-care home relationship require further study. This may include qualitative studies to explore in-depth from various stakeholders what makes a good primary care-care home working relationship and how this can be achieved.

## 5. Conclusions

These results show a shift among GP practices to working with fewer care homes, with fewer accounts of fractured working relationships, within a wider context of various approaches and numbers of care homes that individual practices work with. This could be indicative of improvements in access to primary care. Ways of working that enable access to healthcare that reflect residents′ needs and priorities need to be developed within current structures and resource constraints.

## Figures and Tables

**Table 1 ijerph-19-14774-t001:** Demographic and characteristics of respondents (N = 67).

Characteristics	*n*	%
Age		
18–29	6	8.96
30–39	15	22.38
40–49	21	31.34
50–59	18	26.87
60–69	4	5.97
Not reported	3	4.48
Gender		
Male	28	41.79
Female	37	55.22
Not reported	2	2.98
Ethnicity		
White	45	57.16
Asian	15	22.39
Black	1	1.49
Mixed	1	1.49
Not reported	7	10.45
Job role		
GP	47	70.15
Practice nurse	7	10.45
Nurse practitioner	3	4.48
Practice manager	1	1.49
Other primary care staff	9	13.43
Years worked in primary care		
Less than 2	7	10.45
2–5	9	13.43
5–10	11	16.42
More than 10	40	59.70

**Table 2 ijerph-19-14774-t002:** GP (general practitioner) practice’s current model of care and relationship with care homes.

Variable	*n*	%
Number of care homes		
0	1	1.72
1	17	29.31
2	9	15.52
3	9	15.52
4	10	17.24
5	5	8.62
6	5	8.62
7	1	1.72
15	1	1.72
I do not know	8	12.12
Model of care *		
All care homes are designated same GP	7	7.14
Each care home has its own designated GP	39	39.80
Most of the practice GPs assimilate care home work into everyday caseloads	15	15.31
Locality-led care home-specific teams with expertise in the care of older people	5	5.10
Participate in a Local Enhanced Service for care homes	24	24.49
Other	8	8.16
Relationship with care home *		
Flexible and responsive 2-way communication	28	12.17
Working through informal arrangements as much as formal ones	47	20.43
Supportive and educative working relationship	31	13.48
Good personal relationships (e.g., knowing care home staff names and understanding their roles and vice versa)	54	23.48
Efficient working relationship where pre-visit preparation is done by both the care homes and practice to aid visits and assessments	40	17.39
Good adherence to referral procedures and processes	25	10.87
Limited or lack of a relationship between the practice and the care home	1	0.43
Other	4	1.74

* Respondents could select multiple choices for model of care and relationships with the care home.

## Data Availability

For data inquiries, contact the corresponding author.

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
