# Peer review of "Models of Care and Relationships with Care Homes: Cross-Sectional Survey of English General Practices"

_ijerph, 2022, doi:10.3390/ijerph192214774_

Round 1

Reviewer 1 Report

The study falls short by only quantifying the organizational models. The objective of the study could have been better complemented if the different models had been compared with the quality results in order to be able to affirm which model is more suitable to The Enhanced Health in Care Homes framework for England standards.

We do not know if the Supportive and educational working relationship is more often in care homes with its own designated GP or in care homes visited by Locality-led care home-specific teams.

What informative value does knowing that 25 respondents state that in care homes there is Good adherence to referral procedures and processes? Is it a lot? Is it little?

Specifies in great detail the profile of the respondents to the survey. Does it have any effect on the results of it?

It would be good to compare these results with similar studies, and EHCH standards.  Also if we could know the online survey.

Author Response

Reviewers’ comment

Authors’ response

Reviewer 1

The study falls short by only quantifying the organizational models. The objective of the study could have been better complemented if the different models had been compared with the quality results in order to be able to affirm which model is more suitable to The Enhanced Health in Care Homes framework for England standards.

We added the findings when comparing the models of care with how the relationships were categorised in the Results section. We cannot make any strong claims regarding the suitability of The Enhanced Health in Care Homes framework as the respondents were overwhelmingly positive and few responses with from some models. Only one respondent stated they had limited or lacked a relationship between the practice and care home.

We do not know if the Supportive and educational working relationship is more often in care homes with its own designated GP or in care homes visited by Locality-led care home-specific teams.

We now include the relationships described when care home-specific teams were used. Se the Results section.

What informative value does knowing that 25 respondents state that in care homes there is Good adherence to referral procedures and processes? Is it a lot? Is it little?

Referrals are an important part of the primary care-care home working so we asked about this aspect of their relationship as an indicator of a successful working relationship.

Specifies in great detail the profile of the respondents to the survey. Does it have any effect on the results of it?

We would not expect the respondents’ characteristics to influence their practice’s models of care used or the relationship with care homes so we did not test for an effect.

It would be good to compare these results with similar studies, and EHCH standards.  Also if we could know the online survey.

We have included a fuller discussion and comparison with previous studies in the Introduction and Discussions.

We will include the survey as a supplementary file.

Reviewer 2 Report

Excellent paper.

Can you develop the sentence regarding "future works"? What is the best method for the next investigation? 

Author Response

Reviewers’ comment

Authors’ response

Excellent paper.

Thank you for this positive comment.

Can you develop the sentence regarding "future works"? What is the best method for the next investigation? 

As the word limit for the Abstract is 200 words maximum, we unfortunately cannot go into great detail but this future work and investigations are discussed in the Discussion. 

Reviewer 3 Report

Thank you for the opportunity to review the manuscript “Models of care and relationships with care homes: cross-sectional survey of English general practices” (ijerph-1966105). The authors explored the current models of care and relationships between GP practices and care homes with a semi-structured, cross-sectional online survey. The study must be carefully revised.

In the abstract, please avoid abbreviations or write them out (GP).

Please define a clear research question and, if necessary, formulate sub-questions (at the end of Introduction).  Please go into detail about the research questions and hypotheses in the discussion section.

One of my concerns related to the study continues to be related to the introduction literature. Greater details about previous studies results are needed than currently provided that builds the case for having conducted the current study. This could also strengthen the discussion, as it is quite common to refer to findings from those studies relative to the current study findings in the discussion and conclusions sections.

Also, the implications need to be worked out more clearly.

Author Response

Reviewers’ comment

Authors’ response

Thank you for the opportunity to review the manuscript “Models of care and relationships with care homes: cross-sectional survey of English general practices” (ijerph-1966105). The authors explored the current models of care and relationships between GP practices and care homes with a semi-structured, cross-sectional online survey. The study must be carefully revised.

Thank you for this positive comment. We have carefully revised the manuscript in response to the constructive comments to improve the paper.

In the abstract, please avoid abbreviations or write them out (GP).

Thank you for highlighting this issue. We have removed the abbreviations in the Abstract.

Please define a clear research question and, if necessary, formulate sub-questions (at the end of Introduction).  Please go into detail about the research questions and hypotheses in the discussion section.

We have added research questions at the end of the Introduction, and we have discussed them in the Discussion.

One of my concerns related to the study continues to be related to the introduction literature. Greater details about previous studies results are needed than currently provided that builds the case for having conducted the current study. This could also strengthen the discussion, as it is quite common to refer to findings from those studies relative to the current study findings in the discussion and conclusions sections.

We have added more literature in the Introduction and Discussion with more details of this previous work.

Also, the implications need to be worked out more clearly.

We have added more information in the Discussion about the implications of the findings.

Round 2

Reviewer 1 Report

The paper has been improved enough. Congratulations.

For further research I suggest to explore de perspective of the Care Homes

Reviewer 3 Report

The authors have revised their study so that it can be published.